# `Mixup` to the Random Extreme and Its Performances in Robust Image Classification

## Abstract

We introduce `RandomMix`, an inexpensive yet effective method for data augmentation that combines interpolation-based training and negative weights sampling scheme. Rather than training with a unifying mixup policy for combinations of pairs of examples and their labels, we design a separate mixup rule for pairs of data points. This method naturally combines the previous advantages of previous mixup methods, including `Mixup` Zhang et al. (2017), `CutMix` Yun et al. (2019), but it has its own advantages, including a relatively fast training efficiency without introducing any extra components or parameter, strong robustness performance withstands unseen data distributions. We provide empirical results to demonstrate this method. Our experiments on a range of computer vision benchmark datasets show that `RandomMix` stays comparable to other popular mixup methods on accuracy and outperforms other methods on robustness while showing advantages in efficiency.

## 1 Introduction

Robust machine learning has gained significant attention in recent years due to the challenges posed by distribution shifts and corruptions of data in real-world applications Zhang et al. (2019a); Yang et al. (2022); Vaishnavi et al. (2022); Sriramanan et al. (2022); Chhabra et al. (2022); Dhouib et al. (2020); Wang et al. (2022b). These issues arise when the training and testing data follow different distributions, leading to models that fail to generalize well to new data. Several approaches have been proposed to address this problem, including adversarial training Goodfellow et al. (2015); Madry et al. (2017); Salman et al. (2017); Li et al. (2022); Jia et al. (2022); Herrmann et al. (2022), domain adaptation Ben-David et al. (2007; 2010); Lu et al. (2020); Zheng et al. (2016); Wang et al. (2019a;b) and data augmentation Huang et al. (2020; 2022); Hendryck et al. (2020); Na et al. (2021). Further along the line with data augmentation techniques, numerous works provide theoretical justifications for observed robustness and demonstrate how data augmentation can improve it Hein & Andriushchenko (2017); Gong et al. (2020); Zhang et al. (2020b); Wu et al. (2020); Pinot et al. (2021); Kimura (2020); Couellan (2021).

Data augmentation is a subset of regularization approaches Pham & V.Le (2021); Li et al. (2021); Zhou & Konukoglu (2023), which can be either data-agnostic or data-dependent. Data augmentation entails training a model not only with the original data, but also with additional data that has been properly modified, and this technique has led to state-of-the-art results in various machine learning tasks such as text classification, image recognition, etc Suzuki (2022); Papakipos & Bitton (2022); Chun et al. (2022). Among these data augmentation methods, `Mixup` Zhang et al. (2017; 2021), has been shown as a data-agnostic method to improve the robustness of deep-learning models by creating new training examples through interpolating pairs of existing examples. However, existing `Mixup` variants, such as `AugMix`, `FixBi` Hendryck et al. (2020); Na et al. (2021), often require significant computational resources, making them impractical for use in large-scale applications.

To address this issue, we propose a new variant of `Mixup` called `RandomMix` that effectively enhances the performance of deep-learning models against distribution shifts and corruptions of data while being highly

efficient. Our approach fully exploits the stochastic nature of mixup to create new training examples from various types of corrupted data, allowing models to learn robust representations that generalize well to new situations.

We conduct extensive experiments on standard benchmark datasets (CIFAR10/100, ImageNet) and corrupted datasets (ImageNet-C, ImageNet-P, etc.), demonstrating that `RandomMix` outperforms state-of-the-art mixup methods in robustness metrics while maintaining high training efficiency.

In summary, we believe our work has made the following contributions

- We introduce a `RandomMix` idea that can fully exploit the stochastic nature of `Mixup`, allowing various types of corrupted data to be sampled during training, thus promising better generalization and strong robustness.
- With the help of the ablation study, we analyze why `RandomMix` helps with robustness and generalization.
- With sufficient experiments, we demonstrate the effectiveness of `RandomMix` in multiple benchmarks while outperforming others in efficiency.

## 2 Related Works

### 2.1 Data Augmentation and Robustness

Data Augmentation has helped machine learning models achieve high prediction accuracy over various benchmarks Lim et al. (2019); Ho et al. (2019); Zhong et al. (2020); Buslaev et al. (2020); Ghiasi et al. (2020); Kostrikov et al. (2020). For example, it can be applied to images for classification solutions. It is often necessary when confronted with a massive domain of use like an open world as it improves the robustness of a neural network. Many recent works such as Rebuffi et al. (2021) demonstrate that when combined with model weight averaging, data augmentation can significantly boost robust accuracy. More importantly, data augmentation has helped the model learn robust representations that can be generalized across multiple distributions Zheng et al. (2016); Sun et al. (2018); Huang et al. (2020); Min et al. (2020); Wang et al. (2022a)

Most commonly used data augmentation techniques include random cropping, horizontal flipping Krizhevsky et al. (2017), noise injection Moreno-Barea et al. (2018), rotation, contrast, or texture perturbation of the images Wang et al. (2022a). Recently, a data augmentation method called `AugMix` has been proposed to improve the adversarial robustness of deep networks by regularizing frequency biasHendryck et al. (2020). `AugMix` is characterized by its utilization of simple augmentation operations in concert with a consistency loss which improves model robustness and uncertainty estimates to a certain degree. Our method is complementary to these techniques and can be used in conjunction to further improve robustness performance.

It is also worth mentioning that series literature argues the effectiveness of using data augmentation together with certain regularizations Liang et al. (2018); Kannan et al. (2018); Wu et al. (2019); Guo et al. (2019); Zhang et al. (2019b); Shah et al. (2019). A recent piece of evidence validating an efficient regularization to be squared $\ell_2$ norm Wang et al. (2022a).

### 2.2 Mixup

In this paper, we focus more on the the more lightweight augmentation methods from the mixup family. The mixup family originates from the invention of the original `Mixup` Zhang et al. (2017), which constructs virtual training examples by linearly integrating different samples and their labels during training. Verma et al. (2019) proposed `Manifold Mixup` which extends raw input data mixing to immediately hidden layer output mixing, and `CutMix` Yun et al. (2019) spatially crops a random rectangular area on one image to another to generate a new image. `Puzzle Mix` Kim et al. (2020) suggests a `Mixup` method that adds saliency and local statistics of the input data based on `CutMix`. `CoMix` Kim et al. (2021) introduces a submodel minimization

algorithm that continues to improve on the basis of clipping and saliency. `i-Mix` Lee et al. (2021) has shown to be a simple yet effective domain-agnostic regularization strategy for improving contrastive representation learning. `AlignMix` Venkataramanan et al. (2022) geometrically align two images in the feature space to retain mostly the geometry or pose of one image and the texture of the other, thereby connecting it to style transfer. `SmoothMix` Jeong et al. (2021) trains on convex combinations of samples along the direction of adversarial perturbation for each input and aims to make a tradeoff between robustness and accuracy. `SaliencyMix` Uddin et al. (2021) uses a saliency map to pick a representative image patch and mixes this indicative patch with the target image which allows the model to learn more appropriate representations to make better performance. `TransMix` Chen et al. (2022) mixes labels based on the attention maps of Vision Transformers to bridge the gap between the input and label spaces. However, these methods introduce extra components or parameters, creating a lot of training overheads.

Contrary to previous works, we do not intend to compare test performance with the SOTA models. What we hope instead is to use `RandomMix` to improve the robustness of the model, while having the training efficiency comparable with the original `mixup` Zhang et al. (2017). It is also worth mentioning that this method requires only minor code modifications. Since model robustness is an increasingly important issue in modern machine learning, such a lightweight method can be easily applied to other types of learning including semi-supervised learning Berthelot et al. (2019); Sohn et al. (2020), contrastive learning Verma et al. (2021); Lee et al. (2020), privacy-preserving learning Huang et al. (2021) and learning with fairness constraints Chuang & Mroueh (2021) to enhance model robustness.

| Method | Mixup function |
|---|---|
| Input Mixup | $(1 - \lambda)\mathbf{x_1} + \lambda\mathbf{x_2}$ |
| Manifold mixup | $(1 - \lambda)h\left(\mathbf{x_1}\right) + \lambda h\left(\mathbf{x_2}\right)$ |
| CutMix | $(\mathbb{1} - \mathbb{1}_B) \odot \mathbf{x_1} + \mathbb{1}_B \odot \mathbf{x_2}$ |
| Puzzle Mix | $(1 - z) \odot \Pi_1^\top \mathbf{x_1} + z \odot \Pi_2^\top \mathbf{x_2}$ |
| CoMix | $(g(\mathbb{Z}_1 \odot \mathbf{x_B}), ....g(\mathbb{Z}_{m'} \odot \mathbf{x_B}))$ |
| i-MIX | $AUG(X_1) + (1 - \lambda)AUG(X_2)$ |
| AlignMix | $f_2(Mix_\lambda f_1(x_1), f_1(x_2))$ |
| SaliencyMix | $1 - M \odot x_1 + M \odot x_2$ |
| SmoothMix | $(1 - \lambda)x_1 + \lambda \cdot x_2^{(T)}$ |
| TransMix | $A(M)x_1 + (1 - A(M))x_2$ |
| MatrixMix | $\boldsymbol{\lambda} \odot \mathbf{x_1} + (\mathbb{1} - \boldsymbol{\lambda}) \odot \mathbf{x_2}$ |

Table 1: Summary of various mixup functions. $\boldsymbol{\lambda}$ with bolded format denotes a matrix $\lambda$ denotes a scalar.

## 3 Methods

In this section, we will introduce our method. We begin by defining the necessary notations and providing a concise summary of the baseline mixup methods and then introduce a new method by leveraging the stochastic nature of mixup. We name this method `MatrixMix`. Further, we continue to extend the current mixup approaches to an extreme level to create a more effective and efficient method for robust machine learning, which is the main method we propose in this paper, namely `RandomMix`. In addition to describing our proposed method, we also briefly touch an overview of why mixup methods can improve the robustness of models, drawing on existing theoretical results. This complements the current theoretical understanding of mixup and highlights the additional properties that our method can introduce.

### 3.1 Preliminary

We first introduce our notations. We define $\mathbf{x} \in \mathcal{X}$ to be input data and $\mathbf{y} \in \mathcal{Y}$ be its output label. Let $\mathcal{D}$ be the distribution over $\mathcal{X} \times \mathcal{Y}$. We use $(\mathbf{x}, \mathbf{y})$ to denote samples, $\theta$ to denote the trained model, and $l(\cdot, \cdot)$ to denote a generic loss function. The vanilla `Mixup` training process simply trains the model by linearly interpolating two samples $(\mathbf{x_1}, \mathbf{y_1})$ and $(\mathbf{x_2}, \mathbf{y_2})$. Namely, the new output labels are $\lambda\mathbf{y_1} + (1 - \lambda)\mathbf{y_2}$ while the corresponding input are $\lambda\mathbf{x_1} + (1 - \lambda)\mathbf{x_2}$. The goal of the model is then to satisfy the following optimization problem:

$$\theta = \arg\min_\theta l\left(\theta\left(\lambda\mathbf{x_1} + (1 - \lambda)\mathbf{x_2}\right), \lambda\mathbf{y_1} + (1 - \lambda)\mathbf{y_2}\right) \tag{1}$$

where $\lambda$ is a sampled scalar parameter (usually from Beta distributions) with constraints such as $\lambda \in (0, 1)$. We now briefly describe other variants of this method. `Manifold Mixup` Verma et al. (2019) employs

$\lambda h(\mathbf{x_1}) + (1-\lambda)h(\mathbf{x_2})$ for some hidden representation $h$. `CutMix` Kim et al. (2021) mixes data as $(\mathbb{1} - \mathbb{1}_B) \odot x_1 + \mathbb{1}_B \odot \mathbf{x_2}$ for a binary rectangular mask $\mathbb{1}_B$, where $B = [r_x, r_x + r_w] \times [r_y, r_y + r_h]$ with $\lambda = r_w r_h / WH$ and $\odot$ represents the element-wise product. In other words, $B$ is a randomly chosen rectangle covering $\lambda$ proportion of the input. `Puzzle Mix` Kim et al. (2020) is formulated as $(1-z) \odot \Pi_1^\top \mathbf{x_2} + z \odot \Pi_1^\top \mathbf{x_2}$ , where $z_i$ represents a mask in $[0,1]$ with mixing ratio $\lambda = \frac{1}{n}\sum_i z_i$. $\Pi_1$ and $\Pi_2$ represent $n \times n$ transportation plans of the corresponding data with $n$ dimensions. $\Pi_{ij}$ encodes how much mass moves from location $i$ to $j$ after the transport. `CoMix` Kim et al. (2021) performs mixup on a collection of input data and returns a mixed data set of the form $(g(\mathbb{Z}_1 \odot \mathbf{x_B}), ...., g(\mathbb{Z}_{m'} \odot \mathbf{x_B}))$ where $\mathbf{x}_B \in \mathcal{R}^{m \times n}$ denote the batch of input data in matrix form. The aforementioned methods are summarized in Table 1.

### 3.2 `MatrixMix` method

The original `Mixup` method Zhang et al. (2017) improves the training of classification models by generating synthetic samples through linear interpolation of a pair of training samples and their labels using scalar multiplication. Our proposed method, `MatrixMix`, is similar to the original `Mixup` in that it also uses interpolation to mix sample entries. However, `MatrixMix` employs a distinct mixing policy for each pixel of the input image, in contrast to the original `Mixup`, where all inputs share the same mixing policy $\lambda$. More specifically, for an input data $\mathbf{x} \in \mathbb{R}^{m \times n}$, the mixing policy for our `MatrixMix` method is a matrix $\boldsymbol{\lambda} \in \mathbb{R}^{m \times n}$, where each element of $\boldsymbol{\lambda}$ is independently sampled from a Gaussian distribution with mean 0.5 and variance 1.0. The mixing inputs and outputs of `MatrixMix` are computed as follows:

$$\tilde{\mathbf{x}} = \boldsymbol{\lambda} \odot \mathbf{x_1} + (\mathbb{1} - \boldsymbol{\lambda}) \odot \mathbf{x_2}, \quad \tilde{\mathbf{y}} = avg(\boldsymbol{\lambda})\mathbf{y_1} + avg(\mathbb{1} - \boldsymbol{\lambda})\mathbf{y_2} \tag{2}$$

where $\odot$ denotes the Hadamard product and $avg(\boldsymbol{\lambda})$ takes the average of all the elements of $\boldsymbol{\lambda}$. The corresponding loss is then

$$\ell\left(\theta\left(\boldsymbol{\lambda} \odot \mathbf{x_1} + (\mathbb{1} - \boldsymbol{\lambda}) \odot \mathbf{x_2}\right), \quad avg(\boldsymbol{\lambda})\mathbf{y_1} + avg(\mathbb{1} - \boldsymbol{\lambda})\mathbf{y_2}\right) \tag{3}$$

A concrete description of the algorithm for `MatrixMix` is given in Algorithm 1. Intuitively speaking, `MatrixMix` simply extends the Vanilla `Mixup` to the pixel level. Every pixel is linearly interpolated with a different sampled parameter stored in $\boldsymbol{\lambda}$, in contrast to the original `Mixup` which uses the same $\lambda$ for all the pixels of an image. This provides better regularization for the training and thus makes our `MatrixMix` method more robust to various noises.

### 3.3 Parameter Distribution

We explain our decision to sample the components of $\boldsymbol{\lambda}$ from a normal distribution rather than the more popular choice of a Beta distribution, as used in the vanilla `Mixup`. Our inspiration for this choice comes from recent works that allow for negative weights on samples through interpolation Krueger et al. (2021), since negative coefficients allow us to extrapolate to

---

**Algorithm 1** `MatrixMix` algorithm

**Input**:
Batch image inputs: $\mathbf{I}$;
Batch image labels: $\mathbf{y}$;
Batch size: $N$;
Neural Network model: $\theta$;
**Output**:
1: **function** GETLOSS($\hat{y}, pred, \mathbf{y}, \boldsymbol{\lambda}$)
2:     $loss \leftarrow \boldsymbol{\lambda}_1 \mathcal{L}(pred, \mathbf{y}) + (\mathbb{1} - \boldsymbol{\lambda}_1)\mathcal{L}(pred; \hat{y})$
3:     return loss
4: **for** $t = 1, 2, ..., M$ **do** Compute shuffled image $I_s$ and label $y_S$ by reordering the indexes
5:     $\forall(i,j): \boldsymbol{\lambda}_{i,j} \sim N(0.5, 1)$
6:     $\mathbf{I}_M^2 \leftarrow \boldsymbol{\lambda} \cdot \mathbf{I} + (\mathbb{1} - \boldsymbol{\lambda}) \cdot \mathbf{I}_s$
7:     $\hat{\mathbf{y}} \leftarrow g(\boldsymbol{\lambda})y + g(\mathbb{1} - \boldsymbol{\lambda})\mathbf{y}_s$
8:     $pred \leftarrow f(\mathbf{I}_M^2; \theta)$
9:     $loss \leftarrow GETLOSS(\hat{y}, pred, \mathbf{y}, \boldsymbol{\lambda})$
10:     Update parameters of $\theta$ according to $loss$

---

more extreme variations, which provide deep model robustness and hence out-of-distribution generalization in the challenging case.

More importantly, our comparison experiments suggest that gaussian distribution leads to better performance while using our proposed `RandomMix`, as shown in Appendix A.5.

### 3.4 Robustness of `Mixup`

Previous studies Zhang et al. (2021) show that the reason why mixup can improve the robustness of the model is that minimizing mixup loss is equivalent to approximately minimizing the upper bound of adversarial loss, so it is robust to adversarial attacks. Here in our work, as a side result, we aim to give some insights as to why `Mixup` works in our experiments, especially in the robustness settings from a feature perspective in parallel with existing theoretical understandings such as Carratino et al. (2020); Chidambaram et al. (2020).

To offer the context of our analytical results, we use $\theta$ to denote the model that is being trained on a dataset $(\mathbf{X}, \mathbf{Y})$ where $\mathbf{X} \sim P$ for some distribution $P$. For a fixed loss function $\ell$, the trained model $\theta$ on the original data set as well as the model $\theta_{mix}$ on the mixed data set are defined as follows:

$$\theta = \arg \min_{\theta \in \Theta} \sum_{(\mathbf{x},y) \in (\mathbf{X},\mathbf{Y})_P} \ell(\theta(\mathbf{x}), y), \quad \theta_{mix} = \arg \min_{\theta \in \Theta} \sum_{(\tilde{\mathbf{x}}_{i,j}, \tilde{y}_{i,j})} \ell(\theta(\tilde{\mathbf{x}}), \tilde{y}) \tag{4}$$

Let $\epsilon_P(\theta)$ be the risk associated with $\theta$, the classifier when we do not mix the features.

**Proposition 1.** *Let $c(\theta_{mix})$ and $c(\theta)$ be defined as in Theorem 3.1 of Wang et al. (2022b). Then $c(\theta_{mix}) \leq c(\theta)$ with the assumption that for any sample $\mathbf{x}_i$ if*

*then*
$$\frac{1}{n} \sum_j I[\theta_{mix}(\tilde{\mathbf{x}}_{i,j}) = \tilde{y}_{i,j}] \geq I[\theta(\mathbf{x}_i) = y_i] \tag{5}$$

$$\max_{\mathbf{x}_{\mathcal{A}(f,\tilde{\mathbf{x}}_{i,j})}} |\theta_{mix}(\tilde{\mathbf{x}}_{i,j}) - y_i| < \max_{\mathbf{x}_{\mathcal{A}(f,x_i)}} |\theta(\mathbf{x}_i) - y_i| \tag{6}$$

*Proof.* See Appendix □

These results may appear trivial at first glance, however, the interesting part lies in the fact that the construction of `Mixup` naturally helps the model counter the tendency to learn the non-semantic features under multiple different definitions. For example, `Input Mixup` interpolates at the pixel level and thus naturally encourage the learning of the pixel-wise features such as texture; cut-mix exchanges certain areas of the input image and thus naturally counter the learning of the salient but spurious features (such as the background) and naturally, improve the models' robustness against perturbation such as randomly erasing a certain area.

While these results are for a binary classification problem, we show in the Appendix how they can be easily generalized to multiclass problems. Thus, we can show that $c(\theta_{mix}) \leq c(\theta)$ to conclude that $\theta_{mix}$ has better theoretical guarantees than $\theta$.

### 3.5 From `MatrixMix` to `RandomMix`: Mixup to Its Random Extreme

Since different mixup methods might have different advantages for various types of corrupted data, we continue to bring the mixup to its extremity of being random to incorporate all these advantages. This will let our model learn more corrupted samples during the training phase and thus have a better generalization performance on the test dataset.

More specifically, we use four mixup methods to concatenate a whole batch dataset, then, as a result, for four evenly sized batches of data, we randomly choose which mixup algorithm from the set of (Vanilla images, `Input Mixup`, `CutMix`, and `RandomMix`) with a further modification of `CutMix` to push it further to the random polarity: we extend `CutMix` from a fixed number of patches to exchange between the two images to multiple patches of the images. We will call this the extreme `RandomMix` in the remainder of this paper.

## 4 Experiments

We conducted experiments to evaluate the effectiveness of our proposed `RandomMix` method. We use ResNet-18 He et al. (2015) for the CIFAR10 dataset Krizhevsky (2009) as the base experiment. We run our experiment

for 200 epochs with stochastic gradient descent with a learning rate set to be 0.1 at an initial stage and then decrease through the CosineAnnealingLR technique Loshchilov & Hutter (2017). The batch size is set to 64, and pixels are all normalized to be [0, 1]. All these above experiments are repeated in CIFAR-100 Krizhevsky (2009), a subset of ImageNet Hendrycks et al. (2019) with various models such as WRN28-10 Zagoruyko & Komodakis (2016), ResNest50 Zhang et al. (2020a), and Vision Transformer Alexey et al. (2020) (Tiny Vision Transformer(TinyViT) Wu et al. (2022) for simplicity of training). We first compare our method in robustness metrics with different datasets across different types of corruption in Section 3.5. Then, we briefly touch on the discussion on the performance of accuracy and efficiency that our method can help achieve in Section 3.5 and Section 3.5 respectively. Finally, we perform an ablation study in Section 3.5 to explain why `RandomMix` can achieve a well-balanced between robustness and generalization performance.

| Model Type / Corruption Type | Vanilla | Input | CutMix | Puzzle Mix | CoMix | AugMix | RandomMix |
|---|---|---|---|---|---|---|---|
| WRN-28 / Gaussian Noise | 74.64% | 87.99% | 74.88% | 63.99% | 63.84% | 88.39% | **90.18%** |
| TinyViT / Gaussian Noise | 70.90% | 83.86% | 75.81% | 84.09% | 44.90% | 80.52% | **84.14%** |
| ResNet18 / Random Erase | 61.14% | 69.53% | 61.09% | 59.05% | 69.52% | 62.36% | **71.98%** |
| TinyViT / Random Erase | 77.76% | 86.43% | 80.14% | **89.17%** | 54.29% | 82.48% | 86.90% |
| WRN-28 / Transfer Attack | 26.85% | 35.22% | 24.76% | 22.62% | 35.22% | 36.20% | **40.80%** |
| ResNest50 / Transfer Attack | 56.33% | 67.19% | 63.14% | 49.43% | 46.29% | 60.71% | **73.57%** |

Table 2: Accuracy rates of various `Mixup` methods for background corrupted on CIFAR-10(WRN-28), CIFAR-100(ResNet18), ImageNet-A(ResNesy50 and TinyViT) dataset

### 4.1 Robustness Against Corruption

In order to justify the robustness of `RandomMix` to common data corruption, we first consider adding Gaussian Noise, Random Erasing Zhong et al. (2020), and Transfer Attack Cheng et al. (2017) for verification. Then, we test with two established benchmarks ImageNet-C Hendricks & Dietterich (2019) and ImageNet-P Hendricks & Dietterich (2019).

**Gaussian Noise** In our work, we simply add Gaussian Noise (with mean 0.0, variance 8.0, and amplitude 1.0 as this setting is close to human recognition) on test samples. Table 2 shows comparison results for the test accuracy after the data is perturbed by Gaussian Noise. We perform our experiments on CIFAR-10(WRN-28) and ImageNet-A(TinyViT). From Table 2, we found that test accuracy with `RandomMix` under Gaussian Noise outperforms its counterparts: On CIFAR-10, `RandomMix` decreases only 6.59% after adding Gaussian Noise while others such as `CoMix` decrease by 22.25%. On ImageNet-A, our `RandomMix` decreases by only 2.57% whereas the second best `Puzzle Mix` decreases by 7.1%. All the results mentioned above demonstrate that our method outperforms all the baselines in both generalization performance and corruption accuracy.

**Random Erase** Zhong et al. (2020) selects a rectangle region in an image and erases its pixels with random values. We follow the same setting as Zhong et al. (2020) to use Random Erasing with a probability of 0.5. We set the minimum erasing area at 0.02, the maximum erasing area at 0.4, and min aspect ratio at 0.3. Due to the page limit, we only report the results on CIFAR100 and ImageNet-A dataset for corruption robustness evaluation. Table 2 demonstrates that our method using `RandomMix` improves both the generalization performance and the corruption accuracy by decreasing 7.25% and 4.81% under Random Erase corruption with ResNet18 and TinyViT respectively.

**Transfer Attack** Huang et al. (2019) tests the ability of an attack against a machine-learning model to be effective against a different, potentially unknown, model. It can also be used to disrupt images. Specifically, we first generate attacked images from the source model and transfer them to our trained (target) model. Both the source model and attack model are models from the ResNet series. From Table 2, we observe `RandomMix` also shows the most satisfactory results.

**ImageNet-C** Hendricks & Dietterich (2019) proposed ImageNet-C, which is made by a total of 15 different corruption data of ImageNet test sets. The corruptions are drawn from four main categories including noise, blur, weather, and digital. Each corruption type has five levels of severity or intensity. It is worth noting that

| Network | ImageNet (↑%) | ImageNet-C (↓%) | ImageNet-P (↓%) |
|---------|---------------|-----------------|-----------------|
| Vanilla | 88.62% | 41.45% | 42.59% |
| Input | 89.28% | 32.11% | 45.56% |
| AugMix | 88.19% | 37.13% | 60.21% |
| CutMix | 86.53% | 45.74% | 49.33% |
| Puzzle Mix | **92.47%** | 34.43% | 49.33% |
| CoMix | 64.90% | 70.71% | 114.07% |
| RandomMix | 89.43% | **30.37%** | **42.34%** |

Table 3: Clean test accuracy of ResNest50 models on ImageNet, average Corruption Error values on ImageNet-C (lower values are better) and mean flip probability for ImageNet-P excluding noise perturbations (lower values are better )

| Network | ImageNet (↑%) | ImageNet-C (↓%) | ImageNet-P (↓%) |
|---------|---------------|-----------------|-----------------|
| Vanilla | 83.23% | 51.02% | 73.87% |
| Input | 89.33% | 40.48% | 58.77% |
| AugMix | 89.28% | 39.28% | 79.54% |
| CutMix | 85.14% | 45.43% | 57.05% |
| Puzzle Mix | **91.19%** | 42.37% | 58.17% |
| CoMix | 43.95% | 75.69% | 137.29% |
| RandomMix | 86.71% | **35.43%** | **55.81%** |

Table 4: Clean test accuracy of TinyVisionTransformer models on ImageNet, average Corruption Error values on ImageNet-C (lower values are better) and mean flip probability for ImageNet-P excluding noise

our models are trained only on clean ImageNet images. The ImageNet-C is only used for robustness testing. Table 6 and Table 7 contain standard test metrics which contain average Corruption Error (mCE) values. In our experiment, corrupted test data manifest at five different severity levels as s, from range 1 to 5, for a specific corruption c, the error rate at severity s is $E_{c,s}$. Compute mean corruption error (mCE) involves first computing the unnormalized corruption of a given corruption type (c) by averaging across the 5 severity levels represented as $uCE_c = \Sigma_{s=1}^{5} E_{c,s}$. Then, we average uCEc for all 15 corruption types to compute mCE. For ImageNet-C, mCE values is computed by averaging across all 15 CE values($CE_{GaussianNoise}$, $CE_{ShotNoise}$, . . . , $CE_{Pixelate}$, $CE_{JPEG}$). We consistently observe a better performance on both of our models. Specifically, our method achieves 30.37% mCE on ResNest-50 and 35.43% mCE on TinyViT as shown in Table 3 and Table 4, down from all of the other state of the art methods.

**ImageNet-P** ImageNet-P Hendricks & Dietterich (2019) contains 10 types of common perturbations. Unlike common corruptions, the perturbations are subtly nuanced, spanning fewer pixels within images. Like ImageNet-C, ImageNet-P also contains noise, blur, weather, and digital distortions, but ImageNet-P differs from ImageNet-C in that perturbation sequences are generated from each ImageNet validation image; each sequence contains more than 30 frames, so dataset size and evaluation time are reduced by using only 10 common perturbations. According to Hendricks & Dietterich (2019), the standard metrics for evaluating a model's robustness under these perturbations are mean flip rate (mFR) and mean top-5 distance (mT5D). We report our results in Table 3 and Table 4. Because perturbation robustness is not measured by accuracy but by whether video frame predictions match, we compute Flipping Probability, definitively, we calculate the likelihood that two adjacent frames or two frames with slightly different brightness levels have "flipped" or mismatched predictions for videos with steadily increasing brightness. There are ten different types of perturbations, and the average of these is the mean Flip Probability (mFP). We can normalize by AlexNet's flip probabilities, as we did with ImageNet-C, to obtain the mean Flip Rate (mFR). Since the formulation of mFR is more involved than mCE, we refer the reader to Hendricks & Dietterich (2019) for more information on this metric. Based on our experiment, we demonstrate that RandomMix performs consistently well from ResNest50 to TinyViT, also leading to state-of-the-art results in robustness and uncertainty estimation. Table 8 and 9 in the Appendix provide results for each perturbation type.

**4.2 Generalization Performance and Training Efficiency**

**CIFAR10 and CIFAR100**  We use three network architectures to compare the performance of various mixup methods and train with residual network He et al. (2015): WRN28-10 Zagoruyko & Komodakis (2016) and ResNet18He et al. (2015). We set the same training protocol among all mixup methods, which trains both WRN28-10 and ResNet18 for 200 epochs respectively. Hyperparameter settings are available in Appendix A.1. We reproduce popular mixup baselines and compare baselines with our method under the same experimental settings described above. We denote each mixup method as Vanilla (No mixup), `Input Mixup`(Original Mixup) Zhang et al. (2017), `CutMix` Yun et al. (2019), `Puzzle Mix` Kim et al. (2020), `CoMix` Kim et al. (2021), `AugMix` Hendryck et al. (2020), and `RandomMix` in all experiment tables. All models are trained on a single NVIDIA GTX 2080Ti GPU using Pytorch for 200 epochs on the training set and evaluated on the test set.

In both CIFAR10 and CIFAR100 classification problems, the models trained using `RandomMix` has a comparatively high pure test accuracy than most of their analogues trained with other mixup methods. We trained both ResNet18 and WRN28-10 on CIFAR10 dataset. `RandomMix` outperforms most of the other mixup baselines in terms of pure test accuracy with 96.77% in the WRN28-10 experiment. We further test Vanilla, `Input Mixup`, `CutMix`, `Puzzle Mix`, `CoMix`, `AugMix`, and `RandomMix` on the CIFAR-100 dataset with both ResNet18 Huang et al. (2017) and DenseNet121 for 200 epochs. We observe that `RandomMix` still achieves the best performance in terms of pure test accuracy among baseline mixup methods with 79.32% on ResNet18 and 82.07% on DenseNet121.

**ImageNet-A**  ImageNet-A Hendrycks et al. (2019) contains the failure cases of ImageNet-trained ResNet50 among web images. We performed a performance comparison of several mixup methods on the ImageNet-A dataset. For a detailed implementation, we follow Bahng et al. (2020) to construct a subset of ImageNet Russakovsky et al. (2014) containing 9 super-classes. To concentrate on the impact of textural bias, we also balance the ratios of sub-class photos for each super-class. In the ImageNet-A experiment, we use ResNest-50 and TinyViT to compare the performance. We follow the training protocol in Hendrycks et al. (2019) with some minor modifications. Hyperparameter settings are in the Appendix A.1. We observe that the performance of `RandomMix` is higher than most of the other mixup training methods, which is consistent with the results from previous experiments on CIFAR10 and CIFAR100. Meanwhile, we further testify our method on TinyViT and get a relatively higher test accuracy among all these baseline methods. What worth to be mentioned is that `RandomMix` takes two times less time than some of the other methods like `AugMix`, `Puzzle Mix`, etc. for a single epoch training.

**4.3 Training Efficiency**

While `RandomMixup` may not always achieve the highest training accuracy compared to other mixup methods, it is relatively efficient. This advantage is particularly noticeable on large datasets like ImageNet. To compare the test accuracy and training time cost of all the mixup methods, we trained models for 200 epochs on CIFAR10. Our results show that, while `CoMix` achieves higher accuracy than `RandomMix`, it takes significantly longer to train. Specifically, `CoMix` requires almost three times the training time per epoch compared to `RandomMix`.

We report the time cost for training on ImageNet-A. As shown in Figure 1 (see Appendix A.4), we recorded the time usage for every one-fourth of 200 epochs. From the figure, we observe that some mixup methods, like `CoMix`, have a higher test accuracy in the initial epochs, but it overfits after 1250 minutes of training. On the other hand, `RandomMix` achieves the highest accuracy performance in the first 600 minutes, which is the same as `InputMix`. We also calculate the polygon area formed by the accuracy and time consumption of all mixup methods, as shown in Figure 1. Our `RandomMix` achieves the highest accuracy within the first 140 minutes. It is worth noting that we utilized a single NVIDIA RTX 2080Ti for all mixup methods, except for `CoMix`, which was trained using 16 Intel I9-9980XE CPU cores and 4 NVIDIA RTX 2080Ti GPUs.

### 4.4 Ablation Study

The robustness and efficiency of `RandomMix` stem from the great use of the stochastic nature of mixup and its concatenation. In this section, we verified the combination of `MatrixMix` with one-fourth concatenation(also named as `RandomMix`) is significantly better in its balance between accuracy performance and robustness than the other combinations on CIFAR-100 with ResNet-18.

Table 5 shows that using only `MatrixMix` results in a 14.07% lower test accuracy than one-fourth concatenation, but it demonstrates strong robustness against corruptions. After being corrupted by Gaussian noise, the accuracy decreases by only 5.39%, which is lower than the decrease observed with one-fourth concatenation (28.37%). This suggests that `MatrixMix` has a strong robust performance and good generalization capabilities but at the cost of pure test accuracy. To test this hypothesis further, we concatenate the `InputMix` with `MatrixMix` (one-second concatenation) and obtain a pure accuracy of 75.96%, which is 3.36% lower than one-fourth concatenation. The corruption accuracy against Gaussian noise and random erase attacks is 54.49% and 64.46%, respectively. Going one step further, we perform a one-third concatenation (with `InputMix`, `MatrixMix`, and original images, each consisting of one-third of the batch size) and achieve a pure accuracy of only 1.47% lower than that of one-fourth concatenation. At the bottom of Table 5, we show a `MatrixMix` of one-fourth concatenation with all other mixups, which finalizes our `RandomMix`, strikes a good balance between test accuracy and robustness.

| Methods | Test Accuracy | Gaussian Noise | Random Erase |
|---|---|---|---|
| MatrixMix | 65.25% | 59.86% | 54.01% |
| MatrixMix(1/2) + Input Mix(1/2) | 75.96% | 54.49% | 64.46% |
| MatrixMix(1/3) + Input Mix(1/3) + Images(1/3) | 77.85% | 50.70% | 65.43% |
| MatrixMix(1/4) + Input Mix(1/4) + CutMix(1/4) + Images(1/4)(RandomMix) | 79.23% | 51.85% | 71.98% |

Table 5: Pure accuracy and Corruption Accuracy on CIFAR100 dataset with ResNet18 for 200 epochs training

## 4 Discussion

`RandomMix` is a general framework that can be applied to any CNN model in a plug-and-play manner with minimal computational overhead. Through extensive evaluation, we have demonstrated that `RandomMix` can significantly reduce the generalization error of state-of-the-art models on computer vision benchmark datasets. In addition, `RandomMix` can mitigate the tendency to memorize corrupt labels and reduce sensitivity to adversarial examples, resulting in improved training stability. Most importantly, `RandomMix` achieves these benefits while maintaining high training efficiency.

`RandomMix` also opens several possibilities for further exploration. First, is it possible to use a purely `MatrixMix` without one-fourth concatenation with other variants of mixups? From a previous ablation study, we know while current `RandomMix` has comparably high training effectiveness and accuracy, it has a much lower corrupt accuracy when compared with a pure `MatrixMix`. Second, can we introduce some structural prior to $\lambda$? For example, we can define some rules such as pixels in proximity should have similar weights, pixels of the same property (e.g., color) should have similar weights, pixels of the same block or patch should have similar weights, etc. Third, can we design a mathematical formulation to fully prove what is the best design of $\mathbf{g}$? Currently, we only take the average of all the elements of $\lambda$, this is fairly straightforward yet without a robust mathematical guarantee.

Also, it is worth mentioning that if one only concerned with robustness performances but not clean accuracy, one can choose to use only the new component introduced in Section 3.2 instead of the full method.

## 5 Conclusion

In this paper, we first introduced `MatrixMix` to get a more generic form of mixup in previous mixup methods for robust image classification problems. The `MatrixMix` extends the stochastic nature of `Mixup` to an

extremity to perform vanilla `Mixup` on every pixel level, and we further add more randomness to this method by applying different mixup methods randomly over samples in the same batch, resultant in our proposed method `RandomMix`. As a side note, we also offer a lightweight theoretical discussion built upon existing theory to explain where the robustness is gained through our method.

We conducted experiments on several benchmark datasets including CIFAR-10, CIFAR-100, and ImageNet-A, showing that our method has a strong performance over multiple robustness metrics. Meanwhile, we also demonstrated that `RandomMix` has a comparatively higher training efficiency among the mixup family, and a comparable accuracy among the family.

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

# A  Appendix

## A.1  Hyperparameter Settings

The `RandomMix` training procedure is outlined in Algorithm 1. For fair comparisons, we set different hyper-parameters for each dataset,

**CIFAR-10** & **CIFAR-100** We trained models via stochastic gradient descent (SGD) with an initial learning rate of 0.1 then decayed using Cosine Annealing LR by which the graph of learning rate change with epoch be similar to cos. We set the momentum as 0.9 and add a weight decay of 0.0001. `RandomMix` is concatenated with four components(With original images, original `Mixup`, `RandomMix`, and Revised `CutMix`, each part consists of one-fourth of a whole batch). We set the number of image slices in the revised `CutMix` as 3 whereas the original `CutMix` is composed of only two slices. The mixing distribution parameter for `RandomMix` $\lambda$ is a matrix with the same size as input data sampled from Gaussian Normal distribution with Norm($\mu = 0.5$, $\sigma^2 = 1.0$), Mixing weight parameter for the target label is to take an average of all values in $\lambda$, named $\mathbf{g}(\lambda)$, while Original `Mixup`, `Puzzle Mix` and CoMix all follow the same setting as in original paper. For `CutMix` on CIFAR-10, we set `CutMix` Probability as zero, while on CIFAR-100, the `CutMix` Probability is set as 0.5, which is a combination of half of the `Mixup` with half the original images.

**ImageNet-A** We modify the training protocol in Hyojin et al.(2020) for a training model with 200 epochs. The learning rate starts from 0.025, then decayed using Cosine Annealing LR. We set the momentum as 0.9 and add a weight decay of 0.0001. In addition, our image size is set as 224x224, and the Mixing distribution parameter $\alpha$ is 1.0 each for `Input Mixup`, `CutMix`, `Puzzle Mix`, and `CoMix`, which follows the settings of the original papers. For `CoMix`, we only train with 100 epochs. For the experiment following the experimental setting of `CutMix` Yun et al. (2019), we use the same hyperparameter without the clean input regularization.

## A.2  Additional Test results

### A.2.1  Test results on ImageNet-C and ImageNet-P

| Network | mCE | Noise | | | Blur | | | | Weather | | | | Digital | | | |
|---|---|---|---|---|---|---|---|---|---|---|---|---|---|---|---|---|
| | | Gauss | Shot | Impulse | Defocus | Glass | Motion | Zoom | Snow | Frost | Fog | Bright | Contrast | Elastic | Pixel | JPEG |
| AugMix | 37.13 | 57.90 | 57.05 | 61.75 | 55.80 | 51.29 | 36.33 | 48.51 | 40.01 | 41.64 | **15.94** | 11.04 | **19.88** | 16.54 | 23.35 | 23.13 |
| CoMix | 70.71 | 77.06 | 78.11 | 81.56 | 84.76 | 79.12 | 75.86 | 76.28 | 73.78 | 70.30 | 82.93 | 32.43 | 84.78 | 51.78 | 67.24 | 66.45 |
| Cutmix | 45.74 | 62.39 | 61.06 | 66.24 | 59.92 | 46.91 | 49.04 | 44.59 | 48.10 | 55.70 | 58.59 | 17.95 | 73.08 | 16.66 | 22.18 | 21.68 |
| Input | 32.11 | 40.31 | 40.22 | 40.91 | **44.20** | 36.04 | **37.31** | 36.42 | **33.13** | 32.61 | 32.83 | 10.61 | 57.51 | 13.56 | 18.60 | 20.86 |
| Puzzle Mix | 34.43 | 48.66 | 49.72 | 51.85 | 46.69 | 38.96 | 38.36 | **35.02** | 35.24 | **32.46** | 27.51 | **9.49** | 54.70 | **11.32** | 28.37 | 26.14 |
| Vanilla | 41.45 | 55.22 | 54.50 | 59.21 | 58.24 | 43.42 | 44.45 | 39.90 | 43.57 | 52.97 | 52.39 | 15.50 | 68.96 | 14.02 | 15.91 | 19.92 |
| RandomMix | **30.37** | **26.10** | **24.61** | **24.84** | 48.51 | **34.54** | 37.81 | 37.81 | 35.73 | 38.59 | 35.69 | 16.25 | 52.58 | 15.55 | **15.69** | 19.49 |

Table 6: ImageNet-C results on ResNest-50. Corruption Error(CE) and mCE values for various methods on ImageNet-C. The mCE value is computed by averaging across 15 CE values, which stands for an overall classification error on corrupted test data.

| Network | mCE | Noise | | | Blur | | | | Weather | | | | Digital | | | |
|---|---|---|---|---|---|---|---|---|---|---|---|---|---|---|---|---|
| | | Gauss | Shot | Impulse | Defocus | Glass | Motion | Zoom | Snow | Frost | Fog | Bright | Contrast | Elastic | Pixel | JPEG |
| AugMix | 39.28 | 58.54 | 56.66 | 59.69 | **51.49** | 45.40 | 43.44 | 54.89 | 46.43 | 40.10 | 20.64 | **14.06** | **27.18** | 18.67 | 26.44 | 30.64 |
| CoMix | 75.69 | 79.30 | 80.33 | 81.78 | 85.87 | 80.64 | 79.09 | 80.99 | 74.37 | 76.85 | 86.42 | 54.77 | 86.40 | 63.79 | 67.06 | 65.61 |
| Cutmix | 45.43 | 63.65 | 63.78 | 68.28 | 51.58 | **38.27** | **38.13** | 40.69 | 48.12 | 55.35 | 61.43 | 21.55 | 74.90 | 18.87 | 20.06 | **23.82** |
| Input | 40.48 | 62.03 | 63.27 | 65.04 | 53.98 | 40.04 | 41.41 | 41.74 | **33.77** | **35.07** | 39.28 | 15.01 | 61.36 | **14.67** | 22.55 | 26.77 |
| Puzzle Mix | 42.37 | 60.59 | 60.59 | 65.88 | 56.97 | 46.43 | 45.55 | 44.85 | 36.59 | 37.04 | 42.90 | 14.51 | 63.35 | 15.24 | 29.00 | 30.99 |
| Vanilla | 51.02 | 69.18 | 69.26 | 72.90 | 61.25 | 49.00 | 51.24 | 43.88 | 53.30 | 58.39 | 70.09 | 27.70 | 78.00 | 22.10 | 21.98 | 28.35 |
| RandomMix | **35.43** | **29.50** | **26.21** | **28.41** | 57.80 | 43.74 | 45.70 | 47.99 | 42.65 | 40.97 | 41.00 | 18.04 | 60.66 | 16.50 | **16.33** | 23.95 |

Table 7: ImageNet-C results on TinyViT. Corruption Error(CE) and mCE values for various methods on ImageNet-C. The mCE value is computed by averaging across 15 CE values, which stands for an overall classification error on corrupted test data.

|  |  | Noise | | Blur | | Weather | | Digital | | | |
|---|---|---|---|---|---|---|---|---|---|---|---|
| Network | mFR | Gaussian | Shot | Motion | Zoom | Snow | Bright | Translate | Rotate | Tilt | Scale |
| Vanilla | 73.87 | 96.87 | 121.77 | 47.83 | 32.37 | 52.17 | 45.14 | 69.15 | 70.87 | 30.70 | 127.93 |
| Input | 58.77 | 72.93 | 97.43 | 53.57 | 40.4 | 47.53 | **25.47** | 50.47 | 59.7 | 27.6 | 112.63 |
| AugMix | 79.54 | 104.77 | 119.1 | 75.63 | 65.07 | 69.57 | 32.33 | 68.68 | 79.4 | 35.97 | 144.9 |
| CutMix | 57.05 | 64 | 79.33 | **37.6** | **27.4** | 47.6 | 35.33 | 42.03 | 55.97 | 29.63 | 151.63 |
| Puzzle Mix | 58.17 | 78.23 | 109.77 | 55.77 | 38.43 | 52.3 | 24.33 | **40.2** | **52.67** | 27.63 | **102.33** |
| CoMix | 137.29 | 132.43 | 178.8 | 94.2 | 58.67 | 94.8 | 59.7 | 139.05 | 176.93 | 93.07 | 345.33 |
| RandomMix | **55.81** | **60.93** | **78.67** | 45.73 | 36.97 | **45.33** | 31.3 | 48.27 | 58.8 | **26.07** | 126.07 |

Table 8: ImageNet-P results on TinyViT. The mean flipping rate (mFR) is the average of the flipping rates across all 10 perturbation types. (lower is better)

|  |  | Noise | | Blur | | Weather | | Digital | | | |
|---|---|---|---|---|---|---|---|---|---|---|---|
| Network | mFR | Gaussian | Shot | Motion | Zoom | Snow | Bright | Translate | Rotate | Tilt | Scale |
| Vanilla | 42.59 | **49.47** | 63.7 | 41.27 | **24.43** | 43.23 | **25.2** | 35.8 | 35.9 | **17.93** | 89.63 |
| Input | 45.56 | 60.1 | 63.63 | 53.37 | 38.23 | **37.93** | 21.87 | 32.6 | 43.6 | 21.87 | **82.4** |
| AugMix | 60.21 | 72.23 | 78.93 | 55.00 | 59.8 | 61.5 | 23.73 | 47.25 | 52.1 | 29.83 | 121.67 |
| CutMix | 49.33 | 59.07 | 67 | 55 | 31.5 | 64.05 | 29.33 | 32.43 | 40.13 | 21.03 | 91.03 |
| Puzzle Mix | 49.33 | 59.07 | 67 | 55 | 31.5 | 64.05 | 29.33 | 32.43 | 40.13 | 21.03 | 91.03 |
| CoMix | 114.07 | 138.3 | 161.13 | 135.47 | 65.76 | 111.9 | 53.53 | 92.53 | 121.67 | 69.53 | 190.93 |
| RandomMix | **42.34** | 53.6 | **59.07** | **37.2** | 28.43 | 39.63 | 26.33 | **32.47** | **35.2** | 19.7 | 91.03 |

Table 9: ImageNet-P results on ResNest-50. The mean flipping rate(mFR) is the average of the flipping rates across all 10 perturbation types. (lower is better)

### A.2.2  Test Results with HFC

**High-frequency Component** In order to further validate our `RandomMix` method, we add the HFC test as an extra one. Wang et al. (2019c) proposed a model generalization observation work. The author has pointed out that compared with the shuffled label samples, those natural samples tend to learn more about low-frequency information, and their generalization power on HFC is much lower than on LFC which means that there is a strong connection between marked labels and low-frequency information, thus it's easy for the model to memorize the potential associations. The generalization performance in the shuffled label case is much better on its HFC than LFC, it turns out that the model tends to learn both HFC and LFC information of the given samples. Considering that our methods mix both data and its target labels, we try to use HFC and LFC to prove our assumption that, our model intends to learn both HFC and LFC information. Note that we use identical hyperparameters for both low-frequency and high-frequency radius $r$ for all mixup methods which are provided in Appendix, We take $r$ equals 4, 8, 12,16, 20 on CIFAR10(ResNet18) as the threshold set to align with the HFC paper's settings, `RandomMix` behaves as expected in LFC: it reports a consistently higher prediction accuracy when the radius sustain grow from $r = 4$ to $r = 20$.It suggests that our method can pick more information in LFC compared with other `Mixup` baselines. It also reports a relatively lower prediction accuracy in HFC mode. However, `RandomMix` seems to be more sensitive when $r = 20$ which indicates that due to the trade-off between robustness and accuracy, the model tends to first choose to learn much information while later it turns to choose a generalization pattern from $r = 16$ to $r = 20$. We can conclude that: (1) `RandomMix` can both capture LFC and HFC images during training; (2) when it turns to HFC, `RandomMix` encourages choosing robustness due to the trade-off between test accuracy and generalization performance.

### A.3  Proof of Proposition 1

*Proof.* From Theorem 3.1 of wang2022toward, we have $c(\theta) = \frac{1}{n} \sum_{(x,y) \in (X,Y)_{P_s}} I[\theta(x) = y] r(\theta, \mathcal{A}(f_m, \mathbf{x}))$, where $r(\theta, \mathcal{A}(f, \theta)) = \max_{x_{\mathcal{A}(f, \mathbf{x}_i)}} |\theta(\mathbf{x}_i) - y_j|$, and $c(\theta_{mix}) = \frac{1}{n^2} \sum_{(\tilde{\mathbf{x}}_{i,j}, \tilde{y}_{i,j})} I[\theta_{mix}(\mathbf{x}) = y] r(\theta_{mix}, \mathcal{A}(f_m, \mathbf{x}))$. In this paper, they consider the binary classification problem, but we can easily generalize this to our setting

| Datasets(CIFAR10)Models(ResNet18) HFC(Low Frequency Data) | Vanilla | Input | CutMix | Puzzle Mix | CoMix | RandomMix | AugMix |
|---|---|---|---|---|---|---|---|
| frequency=4 | 16.14% | 14.45% | 18.05% | 16.59% | 13.36% | 13.57% | **27.94%** |
| frequency=8 | 37.07% | 39.46% | 33.34% | 31.70% | 24.11% | 41.35% | **79.20%** |
| frequency=12 | 76.99% | 80.84% | 76.93% | 75.84% | 79.55% | 86.93% | **89.26%** |
| frequency=16 | 88.26% | 90.36% | 88.10% | 88.20% | 91.39% | **93.77%** | 92.29% |
| frequency=20 | 94.31% | **95.31%** | 94.41% | 94.93% | 96.02% | 95.26% | 93.73% |
| HFC(High Frequency Data) | | | | | | | |
| frequency=4 | 19.69% | 10.26% | 17.39% | 17.06% | 10.50% | 24.72% | **30.62%** |
| frequency=8 | 15.84% | 10.63% | 10.85% | 13.39% | 12.32% | **21.60%** | 18.31% |
| frequency=12 | 11.72% | 10.90% | 10.00% | 14.08% | **14.12%** | 11.24% | 12.35% |
| frequency=16 | 10.83% | 9.82% | 10.37% | 11.34% | **10.87%** | 9.17% | 9.95% |
| frequency=20 | 10.45% | 10.17% | 10.63% | **11.10%** | 8.14% | 10.49% | 9.01% |

Table 10: Accuracy rates of various mixup methods for LFC and HFC on CIFAR-10(ResNet18) dataset.

by defining $|\theta(x) - y| = 0$ if $\theta(\mathbf{x})$ and $y$ as the same class, and 1 if they are in different classes. We write $c(\theta)$ as

$$c(\theta) = \frac{1}{n^2} \sum_{i,j=1}^{n} \frac{1}{2} (I[\theta(\mathbf{x}_i) = y_i] r(\theta, \mathcal{A}(f, \mathbf{x}_i)) + I[\theta(\mathbf{x}_j) = y_j] r(\theta, \mathcal{A}(f, \mathbf{x}_j))) \tag{7}$$

Now we can compare this to the corresponding summation for $\theta_{mix}$ (which also has $n^2$ terms). In particular, we pay attention to the summand $r(\theta_{mix}, \mathcal{A}(f_m, \theta_{mix})$ of this term.

$$r(\theta_{mix}, \mathcal{A}(f, \theta_{mix})) \tag{8}$$

$$= \max_{x_{\mathcal{A}(f, \tilde{\mathbf{x}}_{i,j})}} |\theta(\tilde{\mathbf{x}}_{i,j}) - \tilde{y}_{i,j}| \tag{9}$$

$$= \max_{x_{\mathcal{A}(f, \tilde{\mathbf{x}}_{i,j})}} |\theta_{mix}(\tilde{\mathbf{x}}_{i,j}) - (\lambda y_i + (1 - \lambda) y_j)| \tag{10}$$

$$\leq \max_{x_{\mathcal{A}(f, \tilde{\mathbf{x}}_{i,j})}} |\lambda(\theta_{mix}(\tilde{\mathbf{x}}_{i,j}) - y_i) + (1 - \lambda)(\theta_{mix}(\tilde{\mathbf{x}}_{i,j}) - y_j)| \tag{11}$$

$$\leq \max_{x_{\mathcal{A}(f, \tilde{\mathbf{x}}_{i,j})}} \lambda|\theta_{mix}(\tilde{\mathbf{x}}_{i,j}) - y_i| + (1 - \lambda)|\theta_{mix}(\tilde{\mathbf{x}}_{i,j}) - y_j| \tag{12}$$

$$\leq \lambda \max_{x_{\mathcal{A}(f, \tilde{\mathbf{x}}_{i,j})}} |\theta_{mix}(\tilde{\mathbf{x}}_{i,j}) - y_i| + (1 - \lambda) \max_{x_{\mathcal{A}(f, \tilde{\mathbf{x}}_{i,j})}} |\theta_{mix}(\tilde{\mathbf{x}}_{i,j}) - y_j| \tag{13}$$

$$= \max_{x_{\mathcal{A}(f, \tilde{\mathbf{x}}_{i,j})}} |\theta_{mix}(\tilde{\mathbf{x}}_{i,j}) - y_i| \tag{14}$$

The last 2 inequalities follow from the triangle inequality and property of maximums respectively. Thus, we are left to show:

$$\frac{1}{n} \sum_j I[\theta_{mix}(\tilde{\mathbf{x}}_{i,j}) = \tilde{y}_{i,j}] \max_{x_{\mathcal{A}(f, \tilde{\mathbf{x}}_{i,j})}} |\theta_{mix}(\tilde{\mathbf{x}}_{i,j}) - y_i| \leq I[\theta(\mathbf{x}_i) = y_i] \max_{\mathbf{x}_{\mathcal{A}(f, \mathbf{x}_i)}} |\theta(\mathbf{x}_i) - y_i|, \tag{15}$$

which follows straightforwardly based on the assumption. $\square$

## A.4   Training Efficiency validation

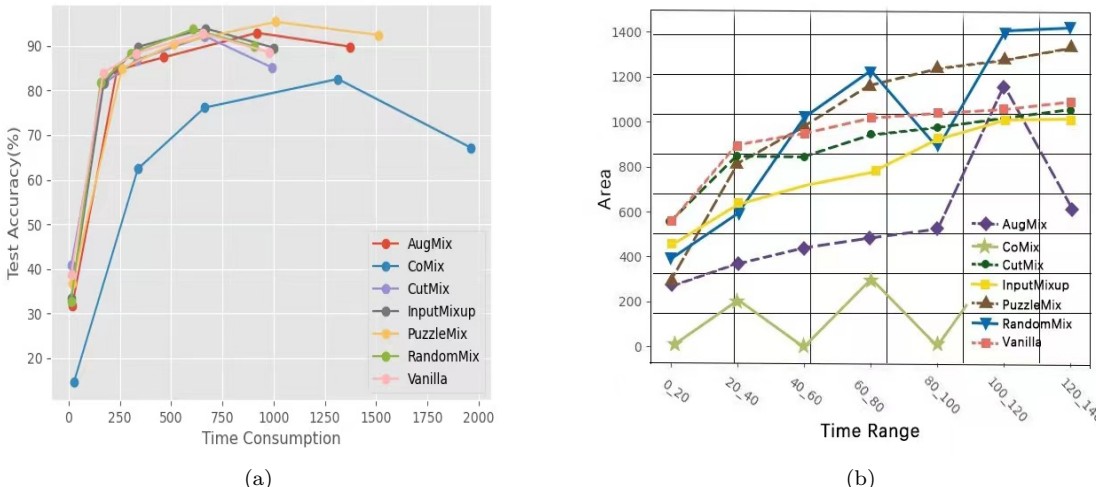

(a)            (b)

Figure 1: (a) Performance of training efficiency of various methods on ImageNet-A(ResNet50). We log the time range for every one-fourth of the total 200 epochs;(b) Area under the curve in figure (a) for the same time periods of various methods on ImageNet-A(ResNet50). It is not a surprise that `RandomMix` decreases in between 80-100mins because there is a relatively slow rise and then after the period, the model breaks through the bottleneck, and there is a rapid rise

| Model\Matrix Mix | CIFAR10/ResNet18 | CIFAR100/ResNet18 | CIFAR10/WRN-28 |
|---|---|---|---|
| Beta Distribution | **92.51%** | **71.54%** | **96.08%** |
| Gaussian(loc=0.5,std=1.0) | 88.57% | 64.18% | 90.70% |
| Model\Random Mix | CIFAR10/ResNet18 | CIFAR100/ResNet18 | CIFAR10/WRN-28 |
| Beta Distribution | 94.82% | **79.67%** | 96.77% |
| Gaussian(loc=0.5, std=1.0) | **95.38%** | 79.56% | **97.07%** |

Table 11: Comparison results for pure accuracy of different models under two distributions with two mixup policies. One is  2, and another is RandomMix.

## A.5   Why choose Gaussian Distribution?

In this work, we use a Gaussian distribution(with the mean set as 0.5, and variance set as 1.0) to replace the original Beta distribution (with both alpha and beta set as 1.0) in Zhang2017Mixup. We do several comparative experiments and report the result in table  11. From the table, the first message we get is that in terms of the `RandomMix` method, pure accuracy is higher with Gaussian Distribution than the one with beta distribution while the situation is reversed in terms of `MatrixMix`. This is probably due to the fact that the central limit theorem states that the sum of many independent random variables is approximately normally distributed, which means that in practice, many complex systems can be successfully modeled as normally distributed noise. Thus, when lacking prior knowledge about the distribution of real numbers under real-world datasets, gaussian distribution is a better choice.

