# OpenReview forum: "Mixup to the Random Extreme and Its Performances in Robust Image Classification"
_TMLR — Rejected by TMLR_

### Review · Reviewer_wAjv · 2023-04-13

**Summary Of Contributions:**

This paper addresses the problem of robustness and how neural networks generalize to good classification accuracy on unseen data distributions (i.e., different datasets). The approach of data augmentation is considered for this purpose. According to the authors, many existing data augmentation techniques require "significant compute." This paper addresses that problem. Table 1 mentions a list of eleven existing and comparable data augmentation techniques, and section 3.2 introduces yet another augmentation. However, the results section seems confusing, as none of the methods are compared in terms of "required compute." Moreover, the results do not compare to the existing literature. Hence, one cannot assess the improvements in terms of robustness.

**Audience:**

Yes

**Broader Impact Concerns:**

This paper seems to address robustness and OOD generalization of neural networks. I would say these topics have a positive impact.

**Claims And Evidence:**

No

**Requested Changes:**




I would suggest to clarify and emphasize the aim of this paper. The current paper addresses two problems but does not answer either. If the aim of the paper were 'reduce required compute at similar-to state-of-the-art results, then I would expect Tables 3, 4, and 5 to report the required compute; if the aim of the paper were 'improve robustness,' then I would expect comparisons to the existing literature in Tables 3, 4, and 5.



Small textual comments:

* What is 'series literature'? See the second paragraph of Related Work

* Section 2.2, 'linearly integrating,' do you mean 'interpolating'?


**Strengths And Weaknesses:**

Strengths:

* The paper is well embedded in the existing literature. Table 1 gives an overview of existing data augmentation techniques, and the other sections have abundant citations.

* Comparisons are run on both ImageNet-C, ImageNet-P, and ImageNet-A.


Weaknesses:

* RandomMix proposes to use different data augmentation at the same time. This idea is as old as InceptionNet [1]. What is novel about this instance?

* Table 3 and Table 4 do not compare to existing literature. How can one be sure that these results are near state-of-the-art? Too often, papers show wins over a weak baseline and results do not hold at the state-of-the-art. For example, the NeurIPS paper [2] achieves 31.40 mCE on Imagenet-C, which is 4 points better than the best result in Table 4. (Likewise, the Resnet50 in CIFAR100 in [3] achieves 7 points accuracy more than the best in Table 5)

* The introduction motivates the paper that existing data augmentation requires 'significant compute.' However,  Tables 3, 4, and 5 do not report the required compute. So how does the method compare in Tables 3, 4, and 5 on required compute?


[1] Szegedy, et al. "Going deeper with convolutions." CVPR. 2015.

[2] Mao, et al. "Enhance the visual representation via discrete adversarial training." NeurIPS 2022

[3] Wightman, Touvron, and Jégou. "Resnet strikes back: An improved training procedure in timm." arXiv 2021

---

### Review · Reviewer_4KTY · 2023-04-20

**Summary Of Contributions:**

They propose a new data augmentation technique, RandomMix, a variant of Mix-up data augmentation that performs the linear interpolation of two images pixel-wise.  They claim that by using the images with their proposed technique, the trained model can get better robustness against image corruption. They perform experiments on the diverse network and datasets and show some improvements on many settings.

**Audience:**

Yes

**Broader Impact Concerns:**

No.

**Claims And Evidence:**

No

**Requested Changes:**

See weaknesses. I think all points are very important to address.

**Strengths And Weaknesses:**

### Strength
1. The proposed augmentation is straightforward to implement.
2. The method demonstrates significant improvement compared to the baseline approaches.
3. The method exhibits a unique approach compared to other existing augmentation techniques.

### Weakness
1. To demonstrate the efficacy of their proposed MatrixMix, the authors should perform an ablation study that compares the results of randomly selecting one augmentation from (Clean, Mixup, CutMix, etc.). This comparison is necessary to showcase how their approach enhances the model's robustness.
2. Although the proposed method enhances the model's robustness to high-frequency noise, its effectiveness on other types of noise is limited due to its specific augmentation design.  Since the proposed method applies mix-up on each pixel independently, the proposed image should include a lot of high-frequency noise. The authors should explicitly mention this limitation in the main paper, rather than just in the appendix, to provide clarity to the readers.
3. The presentation of the proposed method's results may be misleading. Although the results show that the method tailors the model for specific types of corruption, it is crucial to emphasize to the readers that the method does not necessarily make the model generalizable across diverse corruptions. The authors should address this concern and provide additional clarification in the paper.

---

### Review · Reviewer_D1CV · 2023-04-20

**Summary Of Contributions:**

The authors propose a new data augmentation technique called RandomMix. RandomMix randomly selects from no augmentation, Input Mixup, an introduced modification of CutMix, and MatrixMix which is an introduced modification of Mixup. The MatrixMix augmentation is the biggest new introduction in the paper, which applies Mixup on a pixel-by-pixel level, sampling a mixing factor randomly from a Gaussian distribution for each pixel in the image. The authors claim robustness improvements and higher efficiency of RandomMix over previous augmentation techniques, and eval against baselines on ImageNet-C/P and some selected corruptions on CIFAR scale.

**Audience:**

Yes

**Broader Impact Concerns:**

No concerns.

**Claims And Evidence:**

No

**Requested Changes:**


Not all points in the list below might need attention. Some of the requested changes might be able to address by replying to the question or clarifying further.

The the main weaknesses, it would be good to (1) add additional control experience that convincingly rule out the possibility that the observed gain in performance on ImageNet-C is due to the pixelwise augmentation. A feasible way to do this would be to report the mCE by each corruption type.

Also, towards (2) it would be good to clarify and/or rewrite parts of the methods. Happy to discuss the plan here further before taking action, and see my comment above.

Additional suggested changes include (some of these might be able to be resolved by a simple reply within the rebuttal):

1. In Eqs (2) and Eqs. (3), why g to denote the average? just call it $\lambda_\text{avg}$ might be clearer (or sth similar).
2. Can you report the mCE on all corruptions *except* the point-wise noises (Gaussisan, shot noise, etc.)
3. Re 3.4: What is the significance of the theory section 3.4. related to the matrix mixup method? Can you clarify this, either as a reply here or in the paper?
4. Re 3.5: The CutMix extension used here is not clear, can you describe that in more detail as it seems to be a crucial component of the augmentation technique?
5. Re 3.5: Why is matrix mix not part of random mix? I am assuming that RandomMix =(Vanilla images, Input Mixup, CutMix, and MatrixMix), and 3.5 has a typo, please confirm that this is correct.
6. Is a citation missing for TinyViT?
7. Table 2 is somehow hard to read, why not grouping models/corruptions instead of having only single lines?
8. Table 2: What was the rationale behind showing these corruptions in particular? Why not applying all corruptions from e.g. CIFAR-C? In particular the ones not containing gaussian noises are interesteing, gaussian noise seems to be very close to the matrix method
9. What do you mean by "Gaussian Noise attack"? Is it added Gaussian noise, or sth else? Usage of word "attack" might be misleading w.r.t. use in adv. robustness, which this is not, right?
10. "We report the time cost" ... can you put epochs/steps here as well for an impression/comparison?
11. I do not fully understand Table 5: This seems to show that MatrixMix is actually not outperforming the combinations? Can you add MatrixMix to Table 3 and 4 as well, so the individual contribution can be better understood? Can you also add the modified CutMix method to these tables? Also, please separate "own" methods from the baselines you compare to.
12. Re 5: Can you please clarify the sentence "to relax the constraint of convex combination in previous Mixup methods for robust image classification problems" --- which contrained is relaxed with MatrixMix, and how?
13. Re 5: "Meanwhile, we also demonstrated that RandomMix has a comparatively higher training efficiency among the Mixup family" --- where is this shown, exactly?
14. Reference section: Please pay more care in the reference section, some refs are duplicate (e.g. Carratino et al), others lack descriptions of conferences, or arxiv handles, etc., which should be added.
15. Please check the text for typos, mixup vs. Mixup, and for consistency in the method names "MatrixMix", "Matrix", etc.
16. `\citet` and `\citep` are mixed up in the whole paper. Please use `\citet` whenever citations are embedded into the text, and `\citep` for typesetting brackets around citations. This might have happened because the paper was copied from another template and uses \cite everywhere?
17. Sections 4.2 and 4.3 both mention training efficiency, but there is no Figure/table comparing it across the methods. Can you add more quantiative results for this?

**Typos**

1. in 3.3: "that the normal distribution"
2. 3.4.: "naturally penalizeS"
3. 3.5. Isnt it "MatrixMix"?
4. check formatting mCE vs. e.g. mcE
5. "An unnormalized corruption uCEc ..." --- broken sentence
6. 4.1, ImageNet-P: "App." without link to the particular section


**Strengths And Weaknesses:**

**Strengths**

1. The paper is overall well written and easy to follow (although some sections lack context, and some sentences are hard to parse, see below)
2. The literature review is extensive and nicely done, and I liked Table 1 as an overview! However it might be better to have a slighly more verbose caption in Table 1, maybe adding a legend of what the most important symbols within the table mean exactly.
3. There is some empirical evidence that the outlined method improves robustness especially on ImageNet-C and the other considered datasets, but I have concerns over some possible confounders, see below.

**Major Weaknesses**

The central claims of the paper, improved robustness and improved efficiency, seem a bit shaky under the proposed evaluation protocol.

1. The pixelwise mixing technique makes the proposed augmentation **really** similar to the point-wise applied noises in ImageNet-C, including Gaussian Noise. It is known that training on Gaussian noise training improves the robustness on ImageNet-C, and this potentially confounding factor is not addressed at all in the paper. The results by noise type (e.g. "noise", "digital", ...) in the appendix (Table 6 onwards) confirm this, the largest gain comes from the noises. **This cannot be hidden in the main paper, and needs to be discussed.** Other augmentation papers took at least some care to avoid overlap between ImageNet-C corruptions and augmentations. Running on more datasets (like ImageNet-R, ObjectNet, ...) might be helpful in addressing this weakness, see below.
2. The writing style of the paper is a bit confusing in the method section. Right now, the paper introduces MatrixMix in sufficient detail, the CutMix variant is only mentioned in a single sentence, and then these and InputMix are combined to obtain RandomMix. I think the paper needs to make a better job in stating that "RandomMix" is the method, introducing each of its components clearly, and then evaluting it properly. Evaluation should include not only using RandomMix as such, but also including ablations where only CutMix (original) vs. CutMix (adapted) is tested, and MatrixMix is applied to ImageNet-C/P, before everything is finally compiled into RandomMix.

**Minor Weaknesses**

In no particular order, here are additional weaknesses to consider and address.

1. The claim "as negative weights represent inhibitory connections that reduce the output value." is a bit confusing to me. Should this be an analogy to some neural processing mechanism? If so, I do not agree with it, and would drop the term "inhibitory" here/cross out the sentence, I find the connection too speculative and not backed up by a reference, and it is unclear what is meant in the first place.
2. "While RandomMixup may not always achieve the highest training accuracy compared to other mixup methods, it is relatively efficient." --- can you make a table with computaitonal efficiency? I dont get it, shouldnt this be simply the mean of the four employed methods?
3. There is no evaluation on natural distribution shifts like ImageNet-R, ObjectNet, ImageNet-D, etc., although this would be quick to run and does not require re-training.

---

### Decision · Action_Editors · 2023-06-12

**Recommendation:** Reject

**Comment:**

This submission proposes a new mixup-based method for improving robustness of image classification models called RandomMix, which randomly applies one of four augmentation strategies. It shows that RandomMix frequently outperforms previously proposed mixup-based augmentation strategies on a variety of datasets.

Reviewers raised three major concerns regarding the claims made, summarized under "Claims" above. After the rebuttal period, these concerns remain unresolved. Although I am sympathetic to the authors' concerns regarding the cost of performing experiments on ImageNet, the cost to train ResNet-50 on ImageNet is now [fairly reasonable](https://dawn.cs.stanford.edu/benchmark/ImageNet/train.html), and as noted by the reviewers, it is difficult to situate the reported results in the literature without them. I would consider a resubmission that comprehensively addresses all concerns.

**Audience:**

Yes.

**Claims And Evidence:**

There are three major issues with the submission's claims:

- As noted by Reviewers D1CV and 4KTY, the applicability of the proposed method is potentially limited. Evaluations are performed only on synthetically corrupted datasets, where corruptions closely resemble the corruptions produced by randomly mixing images pixelwise. The proposed method works by introducing high-frequency noise, and may not confer robustness in settings where corruption/distribution shift does not primarily involve high frequencies. This concern could be solved by explicitly stating the method's limitations rather than through new experiments.
- ImageNet evaluations are conducted on only 9 classes, and seemingly involve models trained on ImageNet-A. This is a very non-standard evaluation setup not clearly spelled out in the paper, and, as noted by Reviewer D1CV, the results cannot be compared with any other literature. Performing evaluation in a standard setup is important not just for future work that might seek to build upon this paper, but also to verify that the baselines are appropriately implemented, as noted by Reviewer wAjv.
- As noted by Reviewers D1CV and 4KTY, the ablations are insufficient to validate the proposed method. RandomMix uses one of four augmentation strategies at random, two of which are novel (MatrixMix and a novel variant of CutMix). The benefits of these new mixup algorithms over others, either individually or as part of RandomMix, are not explicitly tested.

**Resubmission Of Major Revision:**

The authors may consider submitting a major revision at a later time.